# Rudhira-mediated microtubule stability controls TGFβ signaling during mouse vascular development

**Divyesh Joshi[1†], Preeti Jindal[1†], Ronak K Shetty[1], Maneesha S Inamdar[1,2]***

[1]Jawaharlal Nehru Centre for Advanced Scientific Research, Bangalore, India; [2]Institute for Stem Cell Science and Regenerative Medicine (inStem), Bangalore, India

## eLife Assessment

This **important** work provides another layer of regulatory mechanism for TGF-beta signaling activity. The evidence **convincingly** supports the involvement of microtubules as a reservoir of Smad2/3, and association of Rudhira with microtubules is critical for this process. The work will be of board interest to developmental biologists in general and molecular biologists in the field of growth factor signaling.

**\*For correspondence:**
inamdar@jncasr.ac.in

[†]These authors contributed equally to this work

**Abstract** The transforming growth factor β (TGFβ) signaling pathway is critical for survival, proliferation, and cell migration, and is tightly regulated during cardiovascular development. Smads, key effectors of TGFβ signaling, are sequestered by microtubules (MTs) and need to be released for pathway function. Independently, TGFβ signaling also stabilizes MTs. Molecular details and the in vivo relevance of this cross-regulation remain unclear, understanding which is important in complex biological processes such as cardiovascular development. Here, we use *rudhira/Breast Carcinoma Amplified Sequence 3 (Bcas3)*, an MT-associated, endothelium-restricted, and developmentally essential proto-oncogene, as a pivot to decipher cellular mechanisms in bridging TGFβ signaling and MT stability. We show that Rudhira regulates TGFβ signaling in vivo, during mouse cardiovascular development, and in endothelial cells in culture. Rudhira associates with MTs and is essential for the activation and release of Smad2/3 from MTs. Consequently, Rudhira depletion attenuates Smad2/3-dependent TGFβ signaling, thereby impairing cell migration. Interestingly, Rudhira is also a transcriptional target of Smad2/3-dependent TGFβ signaling essential for TGFβ-induced MT stability. Our study identifies an immediate early physical role and a slower, transcription-dependent role for Rudhira in cytoskeleton-TGFβ signaling crosstalk. These two phases of control could facilitate temporally and spatially restricted targeting of the cytoskeleton and/or TGFβ signaling in vascular development and disease.

## Introduction

Cytoskeletal organization and growth factor signaling are mutually dependent and coordinate multiple cellular processes (*Tzima, 2006*). The multifunctional transforming growth factor β (TGFβ) pathway regulates cell proliferation, differentiation, survival, and migration and is sensitive to cytoskeletal rearrangements (*Tzima, 2006*; *Guo and Wang, 2009*; *Moustakas and Heldin, 2008*). Depletion of TGFβ pathway components causes developmental abnormalities, often leading to embryonic death (*Lebrin et al., 2005*). Ligand binding activates the TGFβ receptor, causing receptor-mediated phosphorylation, followed by nuclear translocation of Smad proteins, resulting in transcriptional regulation of target gene expression (*Hata and Chen, 2016*). Smads, the effectors of the TGFβ pathway, are

competitively bound by several proteins and are a hub for regulation of TGFβ signaling and its intersecting pathways (*Massagué, 2008*). Microtubules (MTs) bind to and sequester Smad2/3 and Smad4, thereby negatively regulating TGFβ signaling, which is relieved upon TGFβ stimulation (*Dong et al., 2000*). Conversely, TGFβ signaling stabilizes MTs (*Gundersen et al., 1994*; *You et al., 2020*) through an as yet unclear mechanism, suggestive of a feedback loop.

During development, TGFβ signaling is activated in endothelial cells (ECs) of the remodeling angiogenic endothelium and is critical for vascular development (*Goumans et al., 2009*). MT stability is also essential for angiogenic sprouting in ECs (*Joshi and Inamdar, 2019*). MTs can be stabilized in multiple ways within the cell, initially by RhoA-mDia actin bundling and thereafter by interaction with multi-protein complexes. Serum components, TGFβ and lysophosphatidic acid (LPA), have been shown to stabilize MTs (*Gundersen et al., 1994*; *Cook et al., 1998*). While LPA stabilizes MTs within 30 min post-stimulation and generically (*Cook et al., 1998*), the effect of TGFβ signaling takes at least 2 hr (*Gundersen et al., 1994*) but is more restricted to and essential for endothelial cell function in vivo. Slow and restricted induction of MT stability by TGFβ as compared to the more generic and rapid function of LPA suggests a requirement for new transcription/translation which may be developmentally important. However, the molecular mechanisms and biological relevance of cytoskeletal interactions with TGFβ signaling remain elusive. Characterization of tissue-restricted regulators of the ubiquitous MT cytoskeleton may aid in defining its specialized context-dependent functions.

In this study, we address the role of the MT-interacting protein Rudhira in connecting TGFβ signaling and MT stability. Rudhira associates with MTs and intermediate filaments (IFs), stabilizes MTs, and promotes directional cell migration (*Joshi and Inamdar, 2019*; *Jain et al., 2012*). Rudhira knockout in mouse causes mid-gestation lethality due to aberrant cardiovascular patterning (*Shetty et al., 2018*). Transcriptome analysis showed that several processes were deregulated in the *rudhira* knockout, especially a large number of TGFβ signaling pathway components (*Shetty et al., 2018*). This suggests that Rudhira may regulate TGFβ signaling for vascular development.

Here, we show that Rudhira-depleted cells have reduced Smad2/3 activation during mouse embryonic development, indicating that Rudhira is a positive regulator of TGFβ signaling. We demonstrate that Rudhira facilitates TGFβ-mediated release of Smad2/3 from MTs. Further, TGFβ signaling stabilizes MTs by inducing transcription, in a Rudhira-dependent manner. Thus, Rudhira bridges TGFβ signaling and MT stability, required for cardiovascular development. Ectopic and aberrant Rudhira expression is seen in several carcinomas and positively correlates with tumor metastatic potential (*Siva et al., 2007*; *Bärlund et al., 2002*; *Gururaj et al., 2006*). Hence, our study will aid in understanding TGFβ signaling and suggest novel ways to regulate it in development and disease.

## Results

### Rudhira depletion attenuates Smad2/3-dependent TGFβ signaling in vivo and in vitro

*Rudhira/Bcas3* is an essential gene, critical for mouse cardiovascular development (*Shetty et al., 2018*). Ubiquitous (*Rudhira*[-/-]) or endothelial (*Rudhira CKO*) knockout of *rudhira* results in angiogenic defects and mid-gestation embryonic lethality (*Shetty et al., 2018*). Transcripts of several components of the TGFβ pathway are deregulated upon Rudhira depletion and the knockout phenocopies mutants of several TGFβ pathway components, whose primary phenotype is in the cardiovascular system (*Goumans et al., 2009*). Hence, we characterized the role of Rudhira in TGFβ signaling. Analysis of the molecular interaction, reaction, and relation networks by KEGG pathway mapping indicated that multiple components of the TGFβ pathway were deregulated in the absence of Rudhira, affecting several processes important for cardiovascular development (*Figure 1A*). To validate this, we used non-silencing (NS) control and *rudhira* knockdown (KD) saphenous vein endothelial cell (SVEC) lines as described before (*Supplementary file 1A and B*; *Shetty et al., 2018*). To test the pathway regulation at the level of receptors, we analyzed transcript and basal as well as active protein levels. While we observed some reduction in transcript levels of *Tgfbr1* and *Tgfbr2* (*Figure 1—figure supplement 2A*), the basal and active protein levels of TGFBR1 remained unchanged in KD cells (, *Figure 1—figure supplement 2*). Interestingly, a constitutively active form of TGFBR1 (TGFBR1T204D), expressed under the CMV promoter, was also suppressed in Rudhira KD (overexpression achieved in NS ~200 folds while in KD ~80 folds) (*Figure 1—figure supplement 2C*). TGFBR1 protein has a short half-life,

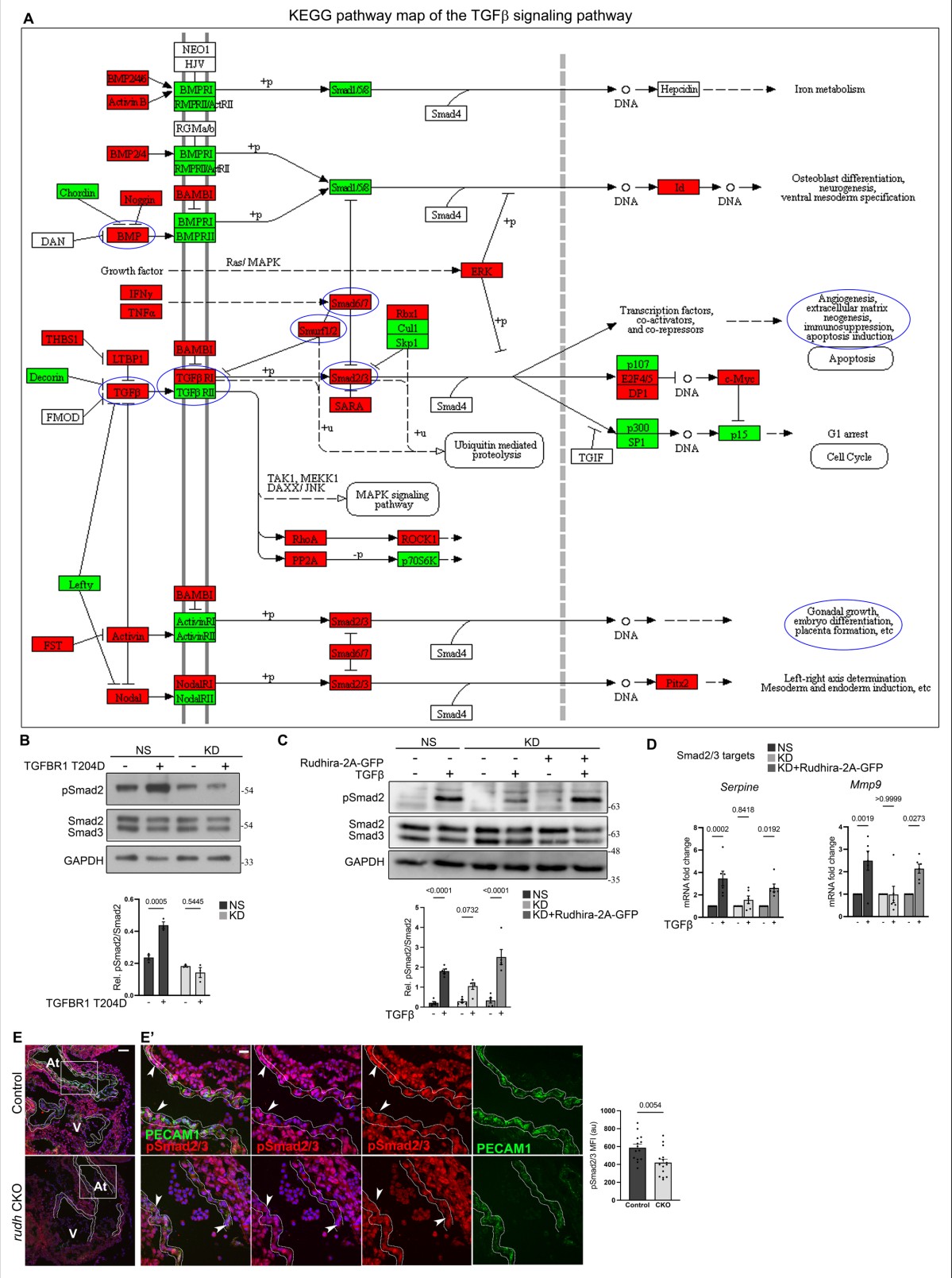

**Figure 1.** Rudhira depletion deregulates developmental endothelial transforming growth factor β (TGFβ) signaling. (**A**) KEGG pathway map indicating TGFβ pathway genes from KEGG database that are deregulated upon *rudhira* depletion in mouse embryonic yolk sac, based on ***Shetty et al., 2018***. Genes were mapped based on fold change in *rudhira*[-/-] yolk sacs in comparison to control to understand pathway regulation. Green: downregulated, red: upregulated. Blue circles indicate affected genes or processes known to be affected upon Rudhira depletion from earlier studies. (**B**) Non-silencing

*Figure 1 continued on next page*

*Figure 1 continued*

(NS) cells and knockdown (KD) saphenous vein endothelial cells (SVECs), transfected with constitutively active TGFBR1 for 24 hr, were analyzed for Smad2 phosphorylation by immunoblotting. Graph shows the quantitation of pSmad2/Smad2 levels (N=3 independent experiments). (**C, D**) Control (NS), *rudhira* KD, and rescued KD (KD+Rudhira-2A-GFP) cells were kept untreated or treated with TGFβ and used for various assays, as indicated. (**C**) Immunoblot for Smad2/3 and pSmad2. Graph shows the quantitation of pSmad2/Smad2 levels (N=3 independent experiments). (**D**) Quantitative RT-PCR (qRT-PCR) analysis of *Mmp9* and *Serpine*, known Smad2/3 targets in endothelial cells (ECs) (N=3 independent experiments). (**E, E′**) Immunostaining for phosphorylated Smad2/3 in control and *rudhira* conditional knockout (*Rudhira CKO*) embryos at E10.5. Boxed region in the embryonic heart in (**E**) marking the endocardium and the myocardium is magnified in (**E′**) (N=3 embryos for each genotype). White dotted line indicates the endocardium. Graph shows the quantitation of pSmad2/3 levels (N=3 independent experiments). Statistical analysis was performed using two-way ANOVA (**A, B, C, D**) and Student's t-test (**E**). Error bars indicate standard error of mean (SEM). *p<0.05, **p<0.01, ***p<0.001. Scale bar: (**E**) 100 µm; (**E′**) 20 µm.

The online version of this article includes the following source data and figure supplement(s) for figure 1:

**Source data 1.** Prism files showing graphs and statistical analysis.

**Source data 2.** Raw uncropped, unedited blots.

**Source data 3.** Uncropped blots with relevant bands labelled.

**Figure supplement 1.** Cell line validation.

**Figure supplement 1—source data 1.** Prism files showing graphs and statistical analysis.

**Figure supplement 1—source data 2.** Raw uncropped, unedited blots.

**Figure supplement 1—source data 3.** Uncropped blots with relevant bands labelled.

**Figure supplement 2.** Rudhira acts downstream to transforming growth factor β (TGFβ) receptors.

**Figure supplement 2—source data 1.** Prism files showing graphs and statistical analysis.

**Figure supplement 2—source data 2.** Raw uncropped, unedited blots.

**Figure supplement 2—source data 3.** Uncropped blots with relevant bands labelled.

**Figure supplement 3.** Rudhira specifically regulates transforming growth factor β (TGFβ)-dependent Smad2/3 activation.

**Figure supplement 3—source data 1.** Prism files showing graphs and statistical analysis.

**Figure supplement 3—source data 2.** Raw uncropped, unedited blots.

**Figure supplement 3—source data 3.** Uncropped blots with relevant bands labelled.

---

while the mRNA is relatively stable (**Vizán et al., 2013**). Reduced amounts of relatively stable RNA may be sufficient to produce adequate protein levels. These increased levels of active TGFBR1 in KD were unable to restore Smad2 phosphorylation, unlike NS controls, which showed a clear increase (**Figure 1B**). While it is important to note that Rudhira or its loss may regulate *Tgfbr* transcript levels, these results support a receptor-independent role of Rudhira in regulating the TGFβ pathway.

To probe further into the molecular mechanism by which Rudhira controls TGFβ signaling, we used *rudhira* knockout mice and the KD EC line. Upon TGFβ stimulation, ECs in culture showed reduced Smad2 activation in the KD as compared to NS control (**Figure 1C**). Further, quantitative RT-PCR (qRT-PCR) analysis showed that transcript levels of Smad2/3 targets, namely *Mmp9* and *Serpine*, did not increase significantly in KD, whereas NS control showed >2-fold increase for both (**Figure 1D**). Expectedly, partial restoration of Rudhira level in KD cells restored Smad2 activation and target gene expression (**Figure 1C and D**). The canonical TGFβ pathway also responds to BMP4 activation. However, upon stimulation with BMP4, there was no significant difference in the levels of pSmad1/5/8 between NS and KD (**Figure 1—figure supplement 2D**). These data indicate that Rudhira is required for TGFβ-dependent Smad2 phosphorylation and pathway activation.

Endothelial depletion of Rudhira (*Rudhira CKO*) is sufficient to cause vascular defects (**Shetty et al., 2018**), and functional TGFβ signaling is essential for vascular development (**Goumans and Mummery, 2000**). Expectedly, *Rudhira CKO* embryos showed a concomitant loss of pSmad2/3 in the embryonic endocardium (**Figure 1E and E′**), without affecting the total Smad2/3 level (**Figure 1—figure supplement 2E, E′**). Thus, Rudhira regulates TGFβ signaling in a Smad2/3-dependent manner in vivo, which is essential for angiogenesis and EC migration during cardiovascular development.

## Rudhira is required for TGFβ-mediated cell migration

As Rudhira is essential for cell migration, we tested the effect of altered TGFβ pathway activity in *rudhira* KD ECs (SVEC) in a transwell migration assay. We also validated our results in HEK293 cells as these allow a high efficiency of transfection and Rudhira overexpression for migration assays. TGFβ

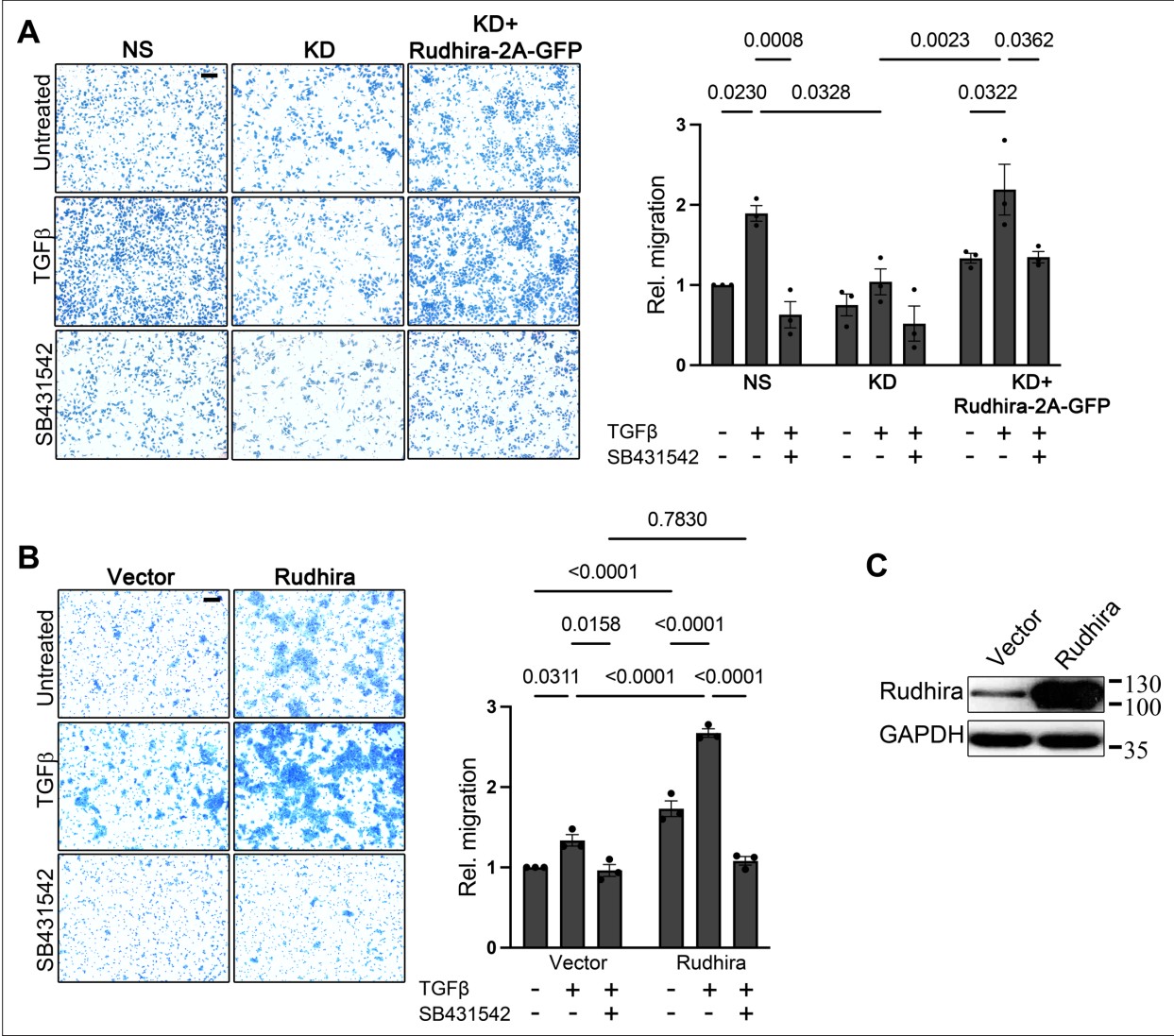

**Figure 2.** Rudhira functions downstream of transforming growth factor β (TGFβ) receptor activation to promote TGFβ-dependent cell migration. (**A**) Control (non-silencing [NS]), *rudhira* knockdown (KD), and rescued KD (KD+Rudhira-2A-GFP) cells were analyzed for migration rates with or without TGFβ using a transwell migration assay. Graph shows the extent of cell migration as measured by crystal violet absorbance (N=3 independent experiments). (**B, C**) HEK293 cells transfected with vector alone or Rudhira were analyzed for migration rates using a transwell migration assay in various conditions, as indicated. Graph shows the extent of cell migration as measured by crystal violet absorbance (N=3 independent experiments). Statistical analysis was performed using one-way ANOVA. Error bars indicate standard error of the mean (SEM). Scale bar: (**A, B**) 100 µm. *p<0.05, **p<0.01, ***p<0.001.

The online version of this article includes the following source data for figure 2:

**Source data 1.** Prism files showing graphs and statistical analysis.

**Source data 2.** Raw uncropped, unedited blots.

**Source data 3.** Uncropped blots with relevant bands labelled.

addition did not enhance migration in *rudhira* KD cells as compared to controls (NS and rescued KD) (*Figure 2A*). In addition, ectopic Rudhira expression in HEK293 cells caused an increase in migration rate as compared to the control, consistent with our earlier study (*Jain et al., 2012*; *Figure 2B and C*).

TGFβ supplementation of Rudhira-overexpressing cells dramatically enhanced the rate of cell migration (~2.6-fold), suggesting a direct correlation between Rudhira levels and pathway function (*Figure 1C and D*). Serum contains TGFβ in a concentration range of 10–40 ng/ml (*Nockowski et al., 2004*), and we used 5% FBS with or without additional TGFβ in the transwell migration assay, since purified TGFβ alone is unable to induce migration. The ratio of increase in migration rates with or

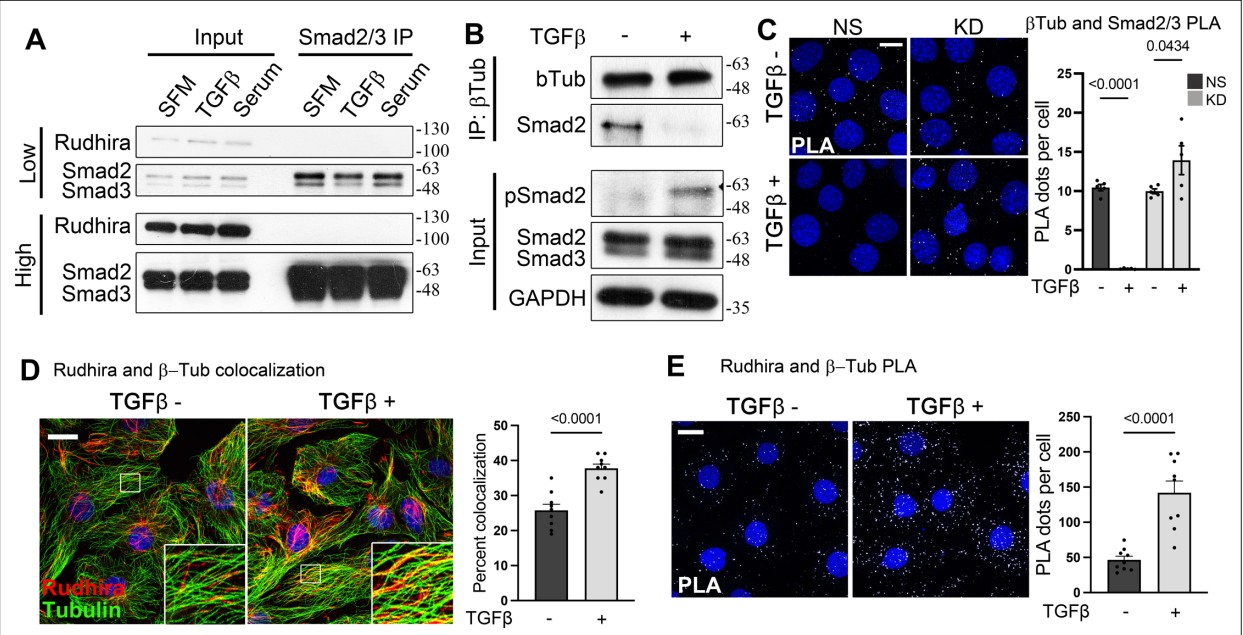

**Figure 3.** Rudhira is essential for the release of Smads from microtubules (MTs) for transforming growth factor β (TGFβ) pathway activation. (**A**) Analysis of interaction between Rudhira, Smad2/3, and MTs by co-immunoprecipitation and immunoblotting, as indicated (N=3 independent experiments). (**B**) Loss of Smad-MT interaction on TGFβ stimulation in saphenous vein endothelial cells (SVECs) (N=3 independent experiments). (**C**) Non-silencing (NS) and knockdown (KD) cells were analyzed for MT and Smad2/3 interaction by proximity ligation assay (PLA) for Tubulin and Smad2/3 with or without TGFβ treatment. PLA dots represent Smad2/3 and β-Tubulin interaction. Graph shows the quantitation of PLA dots per cell (quantitation from >80 cells for two KD cell lines from N=3 independent experiments; also see *Figure 1—figure supplement 3*). (**D, E**) Association of Rudhira and MTs was analyzed by immunostaining (**D**) or PLA (**E**) for Rudhira and Tubulin with or without TGFβ stimulation in SVEC. PLA dots represent Rudhira and β-Tubulin interaction. Graph (**D**) shows the quantitation of colocalization percentage (quantitation from ~40 cells from 9 images each with or without TGFβ, respectively, from N=3 independent experiments) and graph (**E**) shows the quantitation of PLA dots per cell (quantitation from 86 or 68 cells from 9 images each with or without TGFβ, respectively, from N=3 independent experiments). Statistical analysis was performed using one-way ANOVA. Error bars indicate standard error of the mean (SEM). Scale bar: (**C**) 20 μm; (**D, E**) 10 μm. *p<0.05, **p<0.01, ***p<0.001.

The online version of this article includes the following source data for figure 3:

Source data 1. Prism files showing graphs and statistical analysis.

Source data 2. Raw uncropped, unedited blots.

Source data 3. Uncropped blots with relevant bands labelled.

without exogenous TGFβ addition was similar in both the control and Rudhira-overexpressing cells (~1.5-fold). Interestingly, TGFβ pathway inhibition by SB431542 (SB) restored the effect of Rudhira overexpression (enhancement in cell migration rate) to control levels (*Figure 2B*, *Figure 1—figure supplement 3A*), suggesting that Rudhira function is regulated by the TGFβ pathway downstream of receptor activation. In addition to the multiple molecular pathways that function in concert to control cell migration (*Devreotes and Horwitz, 2015*), our data support a role for Rudhira in TGFβ-dependent cell migration.

## Rudhira regulates TGFβ signaling by promoting the release of Smad2/3 from MTs

Since Rudhira localizes to the cytoskeletal components, namely MTs and IFs (*Jain et al., 2012*), we hypothesized that it regulates TGFβ signaling at the cytoskeleton. However, we could not detect any interaction between Rudhira and Smad2/3 under the conditions tested (*Figure 3A*), suggesting that the effect on the pathway may be indirect, possibly through MTs. TGFβ signaling is controlled at multiple levels, including cytoskeletal regulation at MTs, which interact with and sequester Smad2/3, thereby inhibiting the TGFβ pathway (*Dong et al., 2000*). TGFβ stimulation releases Smads from MT sequestration in multiple cell types, including ECs, facilitating its activation and nuclear translocation (*Dong et al., 2000*). Expectedly, TGFβ stimulation in ECs (confirmed by increase in pSmad2 levels)

reduced Smad-MT binding as seen by co-immunoprecipitation for β-Tubulin and immunoblotting for Smad2 (*Figure 3B*).

To assess whether Rudhira is required for TGFβ-dependent release of Smad from MTs, we tested for β-Tubulin and Smad interaction upon TGFβ stimulation in NS and KD cells by performing in situ proximity ligation assay (PLA) (*Bagchi et al., 2015*). While NS cells showed reduced Smad-MT interaction, KD cells retained Smad-MT interaction even upon TGFβ stimulation in multiple KD cell lines (*Figure 3C*, *Figure 1—figure supplement 3B*). Interestingly, TGFβ stimulation resulted in an increase in Rudhira-MT association in wild-type cells, detected by colocalization (*Figure 3D*) and confirmed by PLA (*Figure 3E*). Thus, TGFβ-stimulated Rudhira-MT association makes MTs unavailable for Smad binding. However, MTs are abundant and present as subsets, differing in their Tubulin isotype and isoform composition, posttranslational modifications (PTMs), and stability (*Janke, 2014*), which may also govern their interactions with other molecules. It is unlikely that Rudhira binds to all MTs present. Also, it is likely that only a subset of MTs may bind and sequester Smads.

## *Rudhira* is a transcriptional target of TGFβ signaling and is essential for MT stability

The TGFβ pathway is autoregulatory, due to which several pathway components are induced or inhibited (*Miyazono, 2000*; *Yan et al., 2018*). The *rudhira* promoter harbors putative Smad-binding elements (SBEs) (*Figure 4—figure supplement 1A*). It also harbors putative binding sites of major TGFβ pathway-regulated transcription factors (transcription factor binding sites [TFBS]) (*Figure 4—figure supplement 1B*, *Supplementary file 1c*). Expectedly, TGFβ stimulation increased *rudhira* mRNA and protein levels (*Figure 4A, B, and C*). KD of *Smad2* or *Smad3* resulted in significant reduction of Rudhira levels, indicating that *rudhira* is a Smad2/3-dependent TGFβ target (*Figure 4D*). Blocking transcription with actinomycin D (AcD) or translation using cycloheximide (CHX) prevented the increase in *rudhira* levels upon TGFβ stimulation, confirming that *rudhira* is a transcriptional target of the Smad2/3-dependent TGFβ pathway (*Figure 5A*, *Figure 4—figure supplement 1C*). However, additional investigation is required to confirm whether *rudhira* is a direct target of the Smad2/3-dependent TGFβ pathway.

Rudhira and TGFβ signaling both can induce MT stabilization (*Gundersen et al., 1994*; *Joshi and Inamdar, 2019*). Although the mechanism is unclear, PTMs and/or transcription may control TGFβ-induced MT stability (*Gundersen et al., 1994*). Since in vivo studies indicated that *rudhira* deletion severely affects the TGFβ pathway (*Shetty et al., 2018*), we tested whether TGFβ-induced transcription is required for MT stability. TGFβ stimulation results in a slow enhancement of MT stability over a period of several hours, allowing sufficient time for transcriptional regulation (*Gundersen et al., 1994*). Interestingly, we find that blocking transcription or translation prevented TGFβ-dependent MT stabilization, as detected by Glu-Tubulin levels (*Figure 5A*, *Figure 4—figure supplement 1C*), correlating well with Rudhira levels and validated by increased sensitivity to nocodazole-mediated MT depolymerization (*Figure 5B*). These data show that TGFβ-induced MT stability is transcription-dependent.

## TGFβ signaling stabilizes MTs in a Rudhira-dependent manner

Since *rudhira* transcription is dependent on TGFβ signaling, and Rudhira is required for MT stability, we further tested whether TGFβ-induced MT stability is Rudhira-dependent. Serum starvation resulted in low detectable levels of Glu-Tubulin in NS and KD cells (*Figure 5C*). Stimulation with TGFβ or serum led to increased Glu-Tubulin level in NS but not in KD cells, showing that TGFβ-induced MT stability is dependent on Rudhira (*Figure 5C and D*). To test whether Rudhira also stabilizes MTs in vivo, we analyzed the effect of loss of Rudhira on MT stability in the developing endocardium. Interestingly, along with reduced pSmads levels observed earlier (*Figure 1E and E'*), we found that Glu-Tubulin positive stable MTs were fewer in *Rudhira CKO* embryonic endocardium (*Figure 5D*), supporting a role of Rudhira in regulation of TGFβ signaling and MT stability during cardiovascular development. Combined, our study suggests that Rudhira primarily affects MTs to modulate their Smad-binding property and downstream TGFβ signaling. Consequently, Smad2/3-dependent TGFβ signaling induces *rudhira* transcription, leading to MT stabilization essential for cardiovascular remodeling during development (*Figure 5E*).

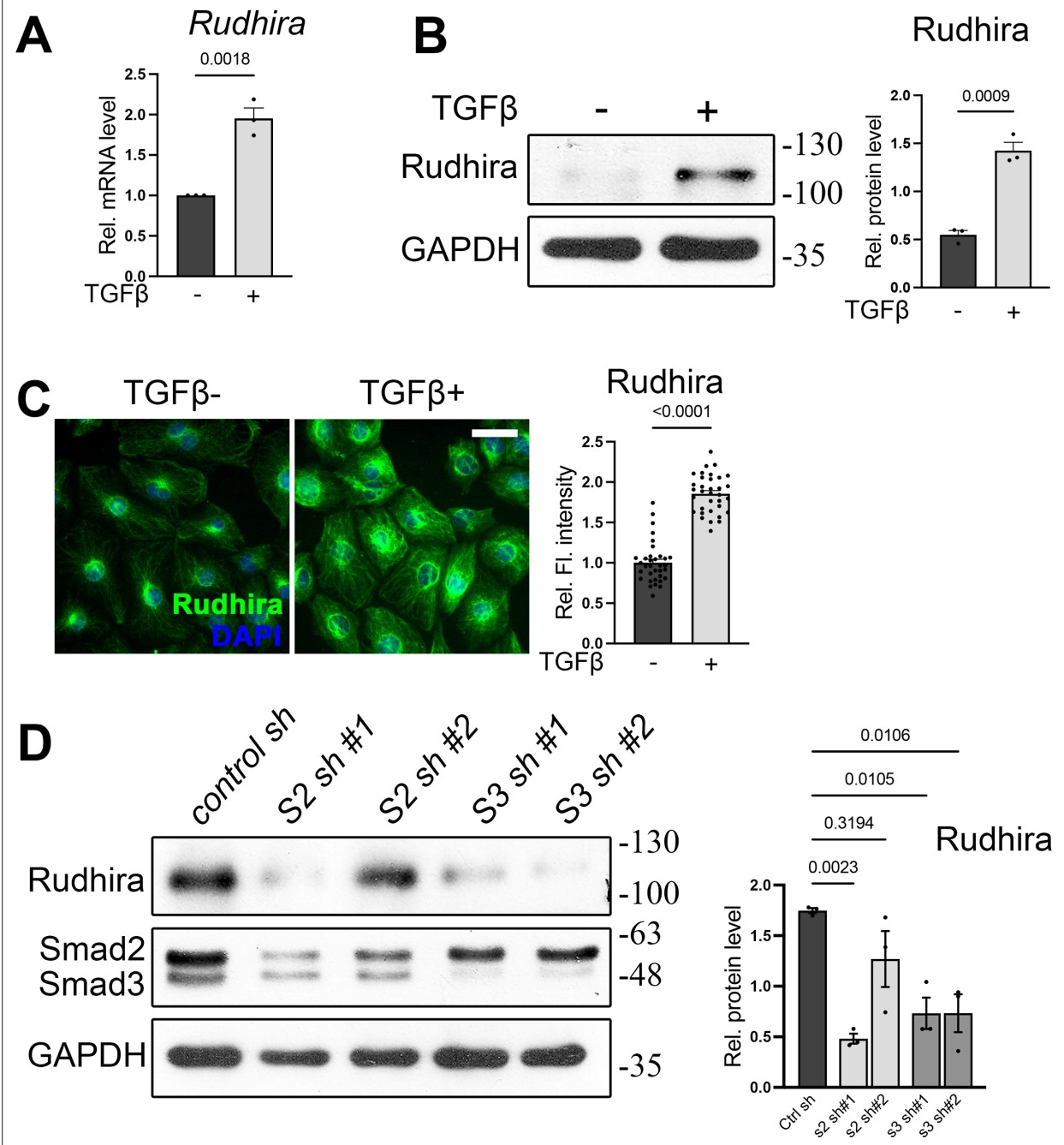

**Figure 4.** Rudhira is a Smad2/3-dependent target of transforming growth factor β (TGFβ) signaling. (**A–C**) TGFβ stimulation followed by quantitative RT-PCR (qRT-PCR) (**A**), immunoblot (**B**), or immunostaining (**C**) analysis (quantitation from 34 cells in each condition) of Rudhira levels in saphenous vein endothelial cells (SVECs). Graphs in (**B**) and (**C**) show the quantitation of Rudhira levels with or without TGFβ (N=3 independent experiments). (**D**) Analysis of Rudhira levels upon *Smad2* or *Smad3* knockdown in HEK293T cells by immunoblotting (N=3 independent experiments). Statistical analysis was performed using one-way ANOVA. Error bars indicate standard error of the mean (SEM). Scale bar: (**C**) 10 μm. *p<0.05, **p<0.01, ***p<0.001.

The online version of this article includes the following source data and figure supplement(s) for figure 4:

**Source data 1.** Prism files showing graphs and statistical analysis.

**Source data 2.** Raw uncropped, unedited blots.

**Source data 3.** Uncropped blots with relevant bands labelled.

**Figure supplement 1.** Transforming growth factor β (TGFβ) induces *rudhira* transcription for microtubule (MT) stability.

**Figure supplement 1—source data 1.** Prism files showing graphs and statistical analysis.

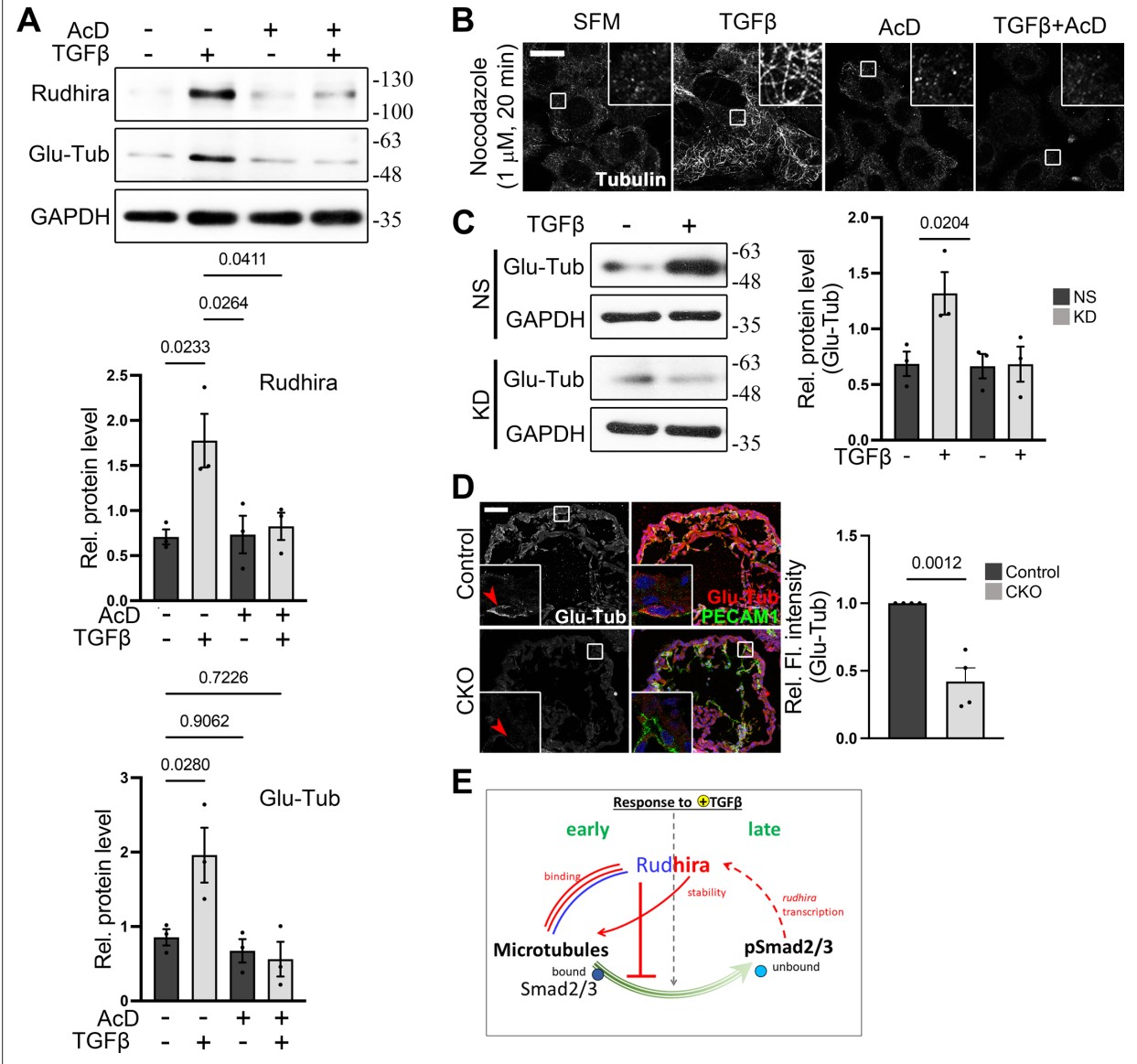

**Figure 5.** Transforming growth factor β (TGFβ)-dependent *rudhira* transcription stabilizes microtubules (MTs). (**A**) Analysis of Rudhira levels and MT stability (marked by Glu-Tubulin) upon treatments as indicated by immunoblotting. Graphs show the quantitation of Rudhira or Glu-Tub levels (N=3 independent experiments). (**B**) Analysis of MT resistance to nocodazole-mediated depolymerization at indicated dosage and time (N=30 cells). (**C**) MT stability in serum-starved non-silencing (NS) cells or knockdown (KD) saphenous vein endothelial cells (SVECs) kept untreated or treated with TGFβ was analyzed by immunoblot for Glu-Tubulin. Graph shows the quantitation of Glu-Tub levels (N=3 independent experiments). (**D**) Immunostaining for Glu-Tubulin in control and *rudhira* conditional knockout (*Rudhira CKO*) embryos at E10.5. Endocardium in the heart is marked by PECAM1. Boxed region in the embryonic heart is magnified in the insets. Red arrowheads in the insets mark the PECAM1 positive cells (N=4 embryos for each genotype). Graph shows the quantitation of Glu-Tub fluorescence intensity in the endocardium. (**E**) Model depicting cellular and molecular response to TGFβ addition. Rudhira has an early and a late effect on MTs. Previously known information about Rudhira is shown in blue. Events identified in this study are shown in red or pink. Parallel lines indicate protein-protein interaction. Solid lines indicate direct effects. Dotted lines indicate effects that may be direct or indirect. Arrows indicate positive regulation. Bar-headed line indicates inhibition. Statistical analysis was performed using one-way ANOVA. Error bars indicate standard error of the mean (SEM). Scale bar: (**B**) 10 µm; (**D**) 100 µm. *p<0.05, **p<0.01, ***p<0.001.

The online version of this article includes the following source data for figure 5:

**Source data 1.** Prism files showing graphs and statistical analysis.

**Source data 2.** Raw uncropped, unedited blots for *Figure 5* and *Figure 4—figure supplement 1*.

**Source data 3.** Uncropped blots with relevant bands labelled for *Figure 5* and *Figure 4—figure supplement 1*.

## Discussion

Growth factor signaling and cytoskeletal remodeling are mutually dependent processes that require complex context-dependent molecular crosstalk (*Tzima, 2006*). Specifically, the developmentally essential, tissue-restricted TGFβ signaling pathway regulates and is regulated by the MT cytoskeleton. However, the underlying molecular mechanism and in vivo relevance of this cross-regulation remain elusive. In this study, we probed the role of Rudhira in the cytoskeletal control of TGFβ signaling and, in turn, its effect on MT properties. Our study identifies a dual role of Rudhira during vascular development - an early role in MT-mediated regulation of Smad2/3 and a late role as a transcriptional target of TGFβ signaling required for MT stability.

While several processes important for cardiovascular development were deregulated in *rudhira*<sup>-/-</sup> embryos, the TGFβ pathway was primarily affected (*Shetty et al., 2018*). TGFβ signaling is a complex pathway and is regulated at multiple levels by various feedback loops. One of the complicated yet delicate feedback circuits is the regulation of receptor levels and activity (*Yan et al., 2018*). Rudhira depletion reduces transcript level without altering basal and active protein level of TGFBR1. This disparity in the transcript and protein level may arise from the fact that while the *Tgfbr* transcript is very stable, the protein is quite unstable, and lower transcript levels may still be sufficient for generating adequate protein levels. Such a phenomenon is not unusual, particularly for membrane proteins, where mRNA levels may not correlate with protein levels. Rudhira, in a manner not yet understood, may also be a part of a broader feedback pathway as it further negatively regulates levels of TGFβ pathway inhibitors such as Smurfs and Smad7 (*Shetty et al., 2018*), suggesting that additional regulatory mechanisms may operate.

TGFβ signaling and Rudhira promote cell migration in a synergistic and interdependent manner. Mechanistically, Rudhira promotes TGFβ-dependent release of Smads from MTs, likely due to increased Rudhira-MT association. This could allow pathway activation and result in enhanced cell migration. Additional, confounding factors downstream of the autoregulatory TGFβ pathway or pathway targets, as described above, may also contribute to Rudhira-dependent cell migration in response to TGFβ. Converse regulation of Rudhira-MT and Smad-MT interaction is indicative of probable competition between Smads and Rudhira for MT binding, which merits further investigation. Binding of Rudhira to MTs may also alter them in a way that makes MTs unavailable for Smad binding. Since TGFβ stimulation is known to stabilize MTs, we hypothesize that TGFβ stimulation increases Rudhira binding to stable MTs. While unclear molecular events involved in the release of Smads from MTs and MT stabilization by Rudhira make it difficult to assign causal roles, this study supports the hypothesis that dissociation of Smads from MTs is essential for Smad activation. In addition, though MTs are ubiquitously and abundantly present, our data suggest that only a few MTs participate in Smad sequestration. This is in concordance with our earlier study, which shows that Rudhira associates preferentially with stable MTs (*Joshi and Inamdar, 2019*).

The TGFβ signaling pathway is autoregulatory, and many of its components are also targets of the pathway (*Miyazono, 2000*). The *rudhira* promoter harbors SBEs. We show that *rudhira*, in addition to being a regulator of the TGFβ pathway, is also a transcriptional target. However, further analysis is required to identify the specific mechanisms by which Smad-dependent, TGFβ-induced, *rudhira* transcription is regulated during development. In contrast to this positive feedback, the TGFβ pathway is also negatively self-regulated, allowing for a return to basal state amenable for reactivation. This self-suppression could possibly involve increased turnover of receptors and Smad2/3, and increased expression of inhibitory Smads, that may recover responsiveness to TGFβ stimulation. Additionally, in the context of MT-dependent Smad2/3 inhibition, the still short turnover time of stable MTs (several minutes to hours) may also promote quick return to resting state. These interesting possibilities can now be tested.

*Rudhira* expression is tightly regulated and restricted to the remodeling endothelium during development (*Shetty et al., 2018*). Mechanistically, Rudhira promotes MT stability for angiogenic remodeling (*Joshi and Inamdar, 2019*). Independent studies show that TGFβ signaling and MT reorganization are essential for cardiovascular development (*Ferrari et al., 2009*; *Martin et al., 2018*; *Bayless and Johnson, 2011*). Deletion of *Smad2/3* or *tubulins* leads to embryonic lethality with severely defective embryonic and extra-embryonic vasculature (*Itoh et al., 2012*). However, the role of MT stability for angiogenic sprouting in vivo and the underlying molecular mechanisms remain elusive. Our study suggests that endothelial cytoskeleton remodeling by Rudhira and activation of

TGFβ signaling during cardiovascular development are interdependent and cross-regulated. TGFβ signaling may lead to restricted induction of MT stability by inducing the expression of MT-stabilizing developmentally regulated proteins like Rudhira. Additional cytoskeletal components such as Vimentin that interacts with Rudhira, and is also a target of TGFβ signaling, are likely to be involved (*Joshi and Inamdar, 2019*; *Jain et al., 2012*). Our analysis will help gain better insight into cellular processes such as migration, which require maintenance of cytoskeletal stability and active signaling for extended periods. An added advantage will be the ability to decipher deregulated migration such as in tumor metastasis. While targeting MT-Smad interaction would allow regulation of the TGFβ pathway, the ubiquitous nature of these molecules is likely to result in widespread and possibly undesirable effects of therapeutic intervention. Rudhira expression, being more restricted, could provide a suitable target to regulate developmental and pathological TGFβ signaling.

## Materials and methods
### Animal experiments
Animal experiments were performed as mentioned earlier (*Shetty et al., 2018*). All animal experimental protocols were approved by the Institutional Animal Ethics Committee (IAEC) of JNCASR (Project number MSI006). All animals were maintained, and experiments were performed according to the guidelines of the animal ethics committee of JNCASR. *Rudhira* floxed mice and conditional knockout mice (*Tek-Cre*-mediated endothelial deletion, *Rudhira^{flox/flox}*; *Tek^{Cre/+}*, abbreviated to *rudhira CKO*) were validated by genotyping. Sample size was estimated based on the previously published papers from our and other labs, using the sample size estimator created by Boston University in compliance with the IACUC guidelines (sample size calculation, bu.edu), considering reasonable variability in means, p-value of less than 0.05 as statistically significant, and the power of the test as 90%.

### Cell culture, transfections, and small molecule treatments
Cell line sources and generation are reported elsewhere (*Shetty et al., 2018*). Identity was authenticated using standard methods of endothelial identification (CD31 staining and diI-AcLDL uptake), and all the cultures were routinely screened for mycoplasma and tested negative. In brief, *rudhira* shRNA vectors (715, 716) and scrambled (NS) control vector (TR30015) (Origene, USA) were microporated into SVEC (mouse EC line) and selected for stable line generation. Control cell line is referred to as 'NS' and *rudhira* knockdown as 'KD'. For Rudhira overexpression and rescue line, the plasmid constructs used were GFP-2A-GFP and Rudh-2A-GFP, which are described elsewhere (*Jain et al., 2012*). Constitutively active TGFBR1 (pcDNA3-ALK5 T204D, 80877) was purchased from Addgene for transfection of SVEC NS and KD cells. Control and shRNA vectors for *Smad2* or *Smad3* knockdown were purchased from the ShRNA Resource Centre (Department of Microbiology and Cell Biology, Indian Institute of Science, Bangalore, India) and were from the SIGMA Library of shRNAs for human gene products. The shRNAs were transfected in HEK293T cells and selected in 1 μg/ml puromycin for 7 days. Details of the oligonucleotide sequence of the shRNAs are provided in *Supplementary file 1a*. HEK293, HEK293T cells were transfected using the calcium phosphate method, while SVECs were transfected using Lipofectamine 2000 (Thermo Fisher Scientific, USA). All small molecules were from Sigma Chemical Co., USA, and treatments were performed in DMEM, unless otherwise indicated. Cells were washed three times in PBS and serum-starved for 12 hr. Thereafter, cells were kept untreated or treated with 10 ng/ml of TGFβ (*Figure 1C*) or BMP4 (*Figure 1—figure supplement 2*) for 2 hr and then taken for immunoblotting. For gene expression analysis, cells were induced with 10 ng/ml of TGFβ for 24–48 hr (*Figures 1D, 4A, B, and C*, *Figure 1—figure supplement 3C*). SB431542 was used at a concentration of 10 μM, as indicated (*Figure 2A and B* and *Figure 1—figure supplement 3A*). For Transwell migration assays, small molecule dilutions were prepared in 5% FBS containing DMEM (*Figure 2A and B*). For interaction studies, cells were serum-starved for 12 hr and thereafter kept untreated or treated with 10 ng/ml of TGFβ or 10% FBS for 2 hr and then taken for PLA, immunostaining, or western blotting, as desired (*Figure 3*, *Figure 1—figure supplement 3A, B*). For TGFβ-dependent *rudhira* transcription and MT stability assays, cells were serum-starved for 48 hr in serum-free medium (SFM) containing 5 mg/ml BSA and 20 mM HEPES (pH 7.4) in DMEM. Thereafter, various treatments were performed for 7 hr (*Figure 5A, B and C*, *Figure 4—figure supplement 1C, D*). TGFβ (10 ng/ml), AcD (10 μM), and CHX (50 μg/ml) were diluted in SFM.

## Quantitative RT-PCR

RNA was isolated using TRIzol reagent (Invitrogen). Reverse transcription was performed using 2 μg of DNase-treated RNA and Superscript II (Invitrogen, Carlsbad, CA, USA) according to the manufacturer's instructions. qRT-PCR was carried out using EvaGreen (Bio-Rad, CA, USA) in Bio-Rad CFX96 Thermal Cycler (Bio-Rad, CA, USA). Details of primers used are provided in *Supplementary file 1b*.

## Immunostaining, immunohistochemistry, fluorescence microscopy, and analysis

Yolk sac or embryos dissected at E10.5 were fixed in 4% paraformaldehyde and processed for cryosectioning (embryos) and immunostaining using standard procedures. Control and knockout embryos were identified by embryonic tail genotyping. Cells were fixed in 4% paraformaldehyde at room temperature or 100% methanol at –20°C and processed for immunostaining using standard procedures (*Schlaeger et al., 1995*). Primary antibodies used were against PECAM1 (CD31) (BD Biosciences, USA), Rudhira, Smad2/3, pSmad2, pSmad2/3 (Cell Signaling Technology), β-Tubulin (DSHB, Iowa), Glu-Tubulin (Abcam), and GFP (Invitrogen). Secondary antibodies were conjugated with Alexa Fluor 488 or Alexa Fluor 568 or Alexa Fluor 633 (Molecular Probes). Bright-field and phase contrast microscopy were performed using an inverted (IX70, Olympus) microscope. Confocal microscope, LSM 880 with Airyscan, Zeiss, was used for fluorescence microscopy. All images in a set were adjusted equally for brightness and contrast using Adobe Photoshop CS2, where required.

## Immunoblot analysis and co-immunoprecipitation

50 μg whole-cell lysate was used for western blot analysis by standard protocols. Blots were cut into strips and incubated with primary antibodies as indicated: Smad2/3, pSmad2, Smad1/5/8, pSmad1/5/8 (Cell Signaling Technology), GAPDH (Sigma Chemical Co., USA), Glu-Tubulin, Ubiquitin, TGFβRI, pTGFβRI (Abcam, USA), β-Tubulin (DSHB, Iowa), Rudhira (Bethyl Labs, USA). HRP-conjugated secondary antibodies against appropriate species were used, and signal developed by using Clarity Western ECL substrate (Bio-Rad, USA). For co-immunoprecipitation, 500 μg whole-cell lysate and 2 μg of the desired antibody were used. Complexes were captured on Protein G-Agarose beads and analyzed by immunoblotting.

## Transwell migration assay

The assays and quantitation were carried out as mentioned earlier (*Jain et al., 2012*). Briefly, 24 hr after transfection with desired plasmid vectors, cells were serum-starved for 12 hr, and 20,000 HEK cells or 40,000 SVECs were plated onto the upper chamber of the transwell filter inserts with 8 μm pore size, 24-well format (Costar, USA). 10% FBS containing medium (with desired small molecules) was added to the lower chamber to serve as a chemoattractant. After 24 hr, cells were fixed in 4% paraformaldehyde for 10 min at room temperature. Cells on the top of the filter were removed using a cotton swab. Cells that had migrated to the bottom were fixed and stained with 0.5% crystal violet for 10 min at room temperature. The dye was extracted in methanol, and absorbance measured spectrophotometrically at 570 nm.

## In situ PLA or Duolink assay

In situ PLA reaction was performed on SVEC lines. The cells were cultured, fixed, permeabilized, and stained with primary antibodies for Smad2/3, β-Tubulin, or Rudhira, as desired. Thereafter, the protocol for PLA as recommended by the manufacturer (Duolink, USA) was followed. Post PLA, nuclei were counterstained with DAPI.

## *Rudhira* promoter in silico analysis

SBEs or motifs and other TFBS were identified from experimental studies, available in JASPAR database (https://jaspar.genereg.net/). PSCAN software (http://159.149.160.88/pscan) was used for motif scanning in mouse and human *rudhira/BCAS3* gene promoters (1 kb sequence upstream of the transcription start site) using matrices obtained from JASPAR bioinformatics tool, as indicated (*Supplementary file 1c*).

## Quantification and statistical analyses

Statistical analyses were performed using one-way ANOVA in the Data Analysis package in Microsoft Excel, and two-way ANOVA and one-sample t-test in GraphPad Prism. $p < 0.05$ was considered significant.

## Materials availability statement

New materials created in this study are freely available upon reasonable request to the corresponding author.

## Acknowledgements

We thank Aksah Sam for maintaining mouse stocks; Developmental Studies Hybridoma Bank, University of Iowa, USA, for some antibodies; JNCASR Imaging Facility, JNCASR Animal Facility, NCBS Animal Facility for access; and Inamdar Laboratory members for fruitful discussions. We are thankful to Arghakusum Das for repeating the experiment for *Figure 4A*. This work was funded by grants from the Department of Biotechnology, Government of India (Sanction no. BT/PR11246/BRB/10/644/2008, dated September 29, 2009), JC Bose fellowship (Grant no. JCB/2019/000020), Department of Science and Technology, Government of India, the Wellcome Trust, UK (094879/B/10/Z), and intramural funds from Jawaharlal Nehru Centre for Advanced Scientific Research, India.

## Additional information

### Competing interests

Maneesha S Inamdar: Reviewing editor, eLife. The other authors declare that no competing interests exist.

### Funding

| Funder | Grant reference number | Author |
|---|---|---|
| Department of Biotechnology, Ministry of Science and Technology, India | Sanction no. BT/PR11246/ BRB/10/644/2008 dated 29.09.2009 | Maneesha S Inamdar |
| JC Bose Fellowship | JCB/2019/000020 | Maneesha S Inamdar |
| Department of Science and Technology, Ministry of Science and Technology, India | | Maneesha S Inamdar |
| Wellcome Trust | 10.35802/094879 | Maneesha S Inamdar |
| Jawaharlal Nehru Centre for Advanced Scientific Research | | Maneesha S Inamdar |

The funders had no role in study design, data collection and interpretation, or the decision to submit the work for publication. For the purpose of Open Access, the authors have applied a CC BY public copyright license to any Author Accepted Manuscript version arising from this submission.

### Author contributions

Divyesh Joshi, Conceptualization, Resources, Data curation, Formal analysis, Validation, Investigation, Visualization, Methodology, Writing – original draft, Project administration, Writing – review and editing; Preeti Jindal, Resources, Data curation, Formal analysis, Validation, Investigation, Visualization, Methodology, Writing – original draft, Writing – review and editing; Ronak K Shetty, Investigation, Methodology; Maneesha S Inamdar, Conceptualization, Resources, Supervision, Funding acquisition, Methodology, Writing – original draft, Project administration, Writing – review and editing

## Author ORCIDs
Divyesh Joshi ⓘ https://orcid.org/0000-0002-4415-2677
Maneesha S Inamdar ⓘ https://orcid.org/0000-0002-8243-2821

## Ethics
All animal experimental protocols were approved by the Institutional Animal Ethics Committee (IAEC) of JNCASR (Project number MSI006). All animals were maintained, and experiments were performed according to the guidelines of the animal ethics committee of JNCASR.

Reviewer #1 (Public review): https://doi.org/10.7554/eLife.98257.4.sa1
Author response https://doi.org/10.7554/eLife.98257.4.sa2

## Additional files

### Supplementary files
Supplementary file 1. ShRNAs, primers, and TF-binding sites in *Rudhira* promoter. (**a**) The oligonucleotide sequence of the *Smad2* and *Smad3* shRNAs. (**b**) Primers used for quantitative RT-PCR (qRT-PCR) analysis. (**c**) Predicted transcription factor binding sites in mouse and human *rudhira/BCAS3* promoters, obtained from JASPAR bioinformatics tool.

MDAR checklist

### Data availability
All data generated or analysed during this study are included in the manuscript and supporting files; source data files have been provided for all figures.

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
