## [Editor Report · eLife Assessment]

This **important** work provides another layer of regulatory mechanism for TGF-beta signaling activity. The evidence **convincingly** supports the involvement of microtubules as a reservoir of Smad2/3, and association of Rudhira with microtubules is critical for this process. The work will be of board interest to developmental biologists in general and molecular biologists in the field of growth factor signaling.

---

## [Referee Report · Reviewer #1 (Public review)]

Summary:

This manuscript aimed to study the role of Rudhira (also known as Breast Carcinoma Amplified Sequence 3), an endothelium-restricted microtubules-associated protein, in regulating of TGFβ signaling. The authors demonstrate that Rudhira is a critical signaling modulator for TGFβ signaling by releasing Smad2/3 from cytoskeletal microtubules and how that Rudhira is a Smad2/3 target gene. Taken together, the authors provide a model of how Rudhira contributes to TGFβ signaling activity to stabilize the microtubules, which is essential for vascular development.

Strengths:

The study used different methods and techniques to achieve aims and support conclusions, such as Gene Ontology analysis, functional analysis in culture, immunostaining analysis, and proximity ligation assay. This study provides unappreciated additional layer of TGFβ signaling activity regulation after ligand-receptor interaction.

Weaknesses:

(1) It is unclear how current findings provide a better understanding of Rudhira KO mice, which the authors published some years ago.

(2) Why do they use HEK cells instead of SVEC cells in Fig 2 and 4 experiments?

(3) A model shown in Fig 5E needs improvement to grasp their findings easily.

---

## [Author Response]

The following is the authors’ response to the previous reviews

According to the reviewers' comments, we appreciate your substantial updates. However, the statistical issue remains unsolved. The following is a general way to get fold changes between controls and experimental samples. Each sample will generate relative differences between target molecules and internal controls. For the case of Fig 1B, the target is pSmad2, and the internal control is the total Smad2. Three control samples will generate three numbers for pSmad2/Smad2 ratios with variations. Similarly, T204D samples will generate three numbers with variations. Then, the average of these three numbers will be set as 1 (with variations) to calculate fold changes between the control and T204D groups. The point is that the statistical significance needs to be evaluated between two groups with variations. This standard method differs from what you described in the manuscript. I hope this explains why the issue needs to be fixed. Please work on the following 11 panels to revise.(1) Fig 1B, WB, pSmad2, reference Smad2, loading control GAPDH, fold change by T204D.(2) Fig 1C, WB, pSmad2, reference Smad2, loading control GAPDH, fold change by Tb/Rudhira.(3) Fig 1D, QRT PCR, pai1/mmp9, fold change by Tb treatment, reference not disclosed.(4) Fig 2A, migration, crystal red absorbance.(5) Fig 2B, migration, crystal red absorbance.(6) Fig 4A, QRT PCR, fold change by Tb.(7) Fig 4B, WB, Rudhira, fold change by Tb.(8) Fig 4C, intensity, with variation, fine.(9) Fig 4D, WB, Rudhira, loading control GAPDH, fold change by Smad2/3 silencing.(10) Fig 5A, WB, Rudhira/Glu-Tub, loading control GAPDH, fold change by Tb and/or AcD.(11) Fig 5C, WB, Glu-Tub.

For western blots:

Graphs for western blots in the following figures have been modified to show the variance in controls, as suggested:

(1) Fig 1B, WB, pSmad2, reference Smad2, loading control GAPDH, fold change by T204D.

(2) Fig 1C, WB, pSmad2, reference Smad2, loading control GAPDH, fold change by Tb/Rudhira.

(7) Fig 4B, WB, Rudhira, fold change by Tb.

(9) Fig 4D, WB, Rudhira, loading control GAPDH, fold change by Smad2/3 silencing.

(10) Fig 5A, WB, Rudhira/Glu-Tub, loading control GAPDH, fold change by Tb and/or AcD.

(11) Fig 5C, WB, Glu-Tub.

For qPCRs:

The reader’s comment asked to display error bars if the variance in controls was considered. The variance in controls was not considered, which is a standard practice in the qPCR assay. In this regard, an example from an eLife paper is cited below (variation not considered in controls):

Fig 4C from Conti et al., N6-methyladenosine in DNA promotes genome stability, revised v2 Feb 3, 2025.

Accordingly, the following graphs remain unchanged:

(3) Fig 1D, QRT PCR, pai1/mmp9, fold change by Tb treatment, reference not disclosed.

(6) Fig 4A, QRT PCR, fold change by Tb.

For crystal violet experiments:

Due to variability in the procedure introduced from CV preparation, uptake, and extraction etc., in the absence of a reference/standard, it is not possible to determine the absolute cell number across experiments. To simplify the calculation, we normalize CV intensity of all the samples to control for an experiment, so the control group doesn’t have error bars. In this regard, an example from an eLife paper is cited below (variation not considered in controls).

Fig 2H from Brunner et al., PTEN and DNA-PK determine sensitivity and recovery in response to WEE1 inhibition in human breast cancer, version of record July 6, 2020.

Accordingly, the following graphs remain unchanged:

(4) Fig 2A, migration, crystal red absorbance.

(5) Fig 2B, migration, crystal red absorbance.

Lastly, #8 remains unchanged.

(8) Fig 4C, intensity, with variation, fine.